**1** **High secondary formation of nitrogen-containing organics (NOCs) and its**

**2** **possible link to oxidized organics and ammonium**

**3** Guohua Zhang[1], Xiufeng Lian[1,2], Yuzhen Fu[1,2], Qinhao Lin[1], Lei Li[3], Wei Song[1], Zhanyong

**4** Wang[4], Mingjin Tang[1], Duohong Chen[5], Xinhui Bi[1,*], Xinming Wang[1], Guoying Sheng[1]

**6** [1] State Key Laboratory of Organic Geochemistry and Guangdong Provincial Key Laboratory of

**7** Environmental Protection and Resources Utilization, Guangzhou Institute of Geochemistry,

**8** Chinese Academy of Sciences, Guangzhou 510640, PR China

**9** [2] University of Chinese Academy of Sciences, Beijing 100039, PR China

**10** [3] Institute of Mass Spectrometry and Atmospheric Environment, Guangdong Provincial

**11** Engineering Research Center for On-line Source Apportionment System of Air Pollution, Jinan

**12** University, Guangzhou 510632, China

**13** [4] School of Intelligent Systems Engineering, Sun Yat-sen University, Shenzhen 518107, PR

**14** China

**15** [5] State Environmental Protection Key Laboratory of Regional Air Quality Monitoring,

**16** Guangdong Environmental Monitoring Center, Guangzhou 510308, PR China

**18** Correspondence to: Xinhui Bi (bixh@gig.ac.cn)

**Highlights**

- Nitrogen-containing organics (NOCs) were highly internally mixed with photochemically produced secondary oxidized organics

- NOCs could be well predicted by the variations of these oxidized organics and ammonium

- Higher relative humidity and NOx may facilitate the conversion of these oxidized organics to NOCs

**Abstract**
Nitrogen-containing organic compounds (NOCs) substantially contribute to light-
absorbing organic aerosols, although the atmospheric processes responsible for the secondary
formation of these compounds are poorly understood. In this study, seasonal atmospheric
processing of NOCs was investigated by single-particle mass spectrometry in urban Guangzhou
from 2013-2014. The relative abundance of NOCs is found to be strongly enhanced when
internally mixed with the photochemically produced secondary oxidized organics (i.e., formate,
acetate, pyruvate, methylglyoxal, glyoxylate, oxalate, malonate, and succinate) and ammonium.
Besides, both the hourly detected particle number and relative abundance of NOCs are highly
correlated with those of secondary oxidized organics and ammonium. It is therefore
hypothesized that secondary formation of NOCs most likely links to the oxidized organics and
ammonium. Results from both multiple linear regression analysis and positive matrix
factorization analysis further show that the relative abundance of NOCs could be well predicted
($R^2 > 0.7$, $p < 0.01$) by the oxidized organics and ammonium.
Interestingly, the relative abundance of NOCs is inversely correlated with ammonium,
whereas their number fractions are positively correlated. This result suggests that although the
formation of NOCs does require the involvement of $NH_3/NH_4^+$ , the relative amount of
ammonium may have a negative effect. Higher humidity and NOx likely facilitate the
conversion of oxidized organics to NOCs. Due to the relatively high oxidized organics and
$NH_3/NH_4^+$, the relative contributions of NOCs in summer and autumn were higher than those in
spring and winter. To the best of our knowledge, this is the first direct field observation study
reporting a close association between NOCs and both oxidized organics and ammonium. These
findings have substantial implications for the role of ammonium in the atmosphere, particularly
in models that predict the evolution and deposition of NOCs.

**Keywords**: nitrogen-containing organic compounds, individual particles, oxidized organics,
ammonium, mixing state, single-particle mass spectrometry

**1 Introduction**

Organic aerosols that strongly absorb solar radiation are referred to as brown carbon (BrC), capable of a comparable level of light absorption in the spectral range of near-ultraviolet (UV) light as black carbon (Andreae and Gelencser, 2006; Feng et al., 2013; Yan et al., 2018). Nitrogen-containing organic compounds (NOCs) substantially contribute to the pool of BrC (Feng et al., 2013; Mohr et al., 2013; Li et al., 2019), and have a significant effect on atmospheric chemistry, human health and climate forcing (Noziere et al., 2015; Kanakidou et al., 2005; Shrivastava et al., 2017; De Gouw and Jimenez, 2009). The particulate organic nitrogen accounts for a large fraction of total airborne nitrogen (~30%), although the proportion exhibits a high variability temporally and spatially, and therefore has an influence on both regional and global N deposition (Neff et al., 2002; Shi et al., 2010; Cape et al., 2011). However, the sources, evolution, and optical properties of NOCs remain unclear and contribute significantly to uncertainties in the estimation of their impacts on the environment and climate (Laskin et al., 2015; Feng et al., 2013).

NOCs are ubiquitous components of atmospheric aerosols, cloud water and rainwater (Altieri et al., 2009; Desyaterik et al., 2013; Laskin et al., 2015), spanning a wide range of molecular weights, structures and light absorption properties (Lin et al., 2016). Emissions of primary NOCs have been attributed to biomass burning, coal combustion, vehicle emissions, biogenic production and soil dust (Laskin et al., 2009; Desyaterik et al., 2013; Sun et al., 2017; Mace et al., 2003; Rastogi et al., 2011; Wang et al., 2017). A growing body of evidence

from laboratory studies suggests that secondary NOCs may be produced in gas phase,
aerosol, and clouds. Maillard reactions involving mixtures of atmospheric aldehydes (e.g.,
methylglyoxal/glyoxal) and ammonium/amines are of particular interest (e.g., Hawkins et
al., 2016; De Haan et al., 2017; De Haan et al., 2011). A significant portion of NOCs may
also be derived from the heterogeneous ageing of secondary organic aerosol (SOA) with
$NH_3$ / $NH_4^+$ (Liu et al., 2015; Laskin et al., 2015). Mang et al. (2008) proposed that even
trace levels of ammonia may be sufficient to form NOCs via this pathway. In addition, gas-
phase formation of NOCs through interaction between volatile organic hydrocarbons and
$NO_x$ and other oxidations, followed by condensation, may have a potential contribution (Fry
et al., 2014; Lehtipalo et al., 2018).
The secondary formation of NOCs is especially prevalent in environments experiencing
high anthropogenic emissions (Yu et al., 2017; Ho et al., 2015), although further studies are
required to establish the formation mechanisms comprehensively. A major obstacle is that
organic and inorganic matrix effects have a profound impact on the chemistry of organic
compounds in bulk aqueous particles and particles undergoing drying (El-Sayed et al., 2015;
Lee et al., 2013). While real-time characterization studies remain a challenge due to the
extremely complex chemical nature of NOCs, establishing this data along with the co-
variation of NOCs with other chemical components would help to identify the sources and
evolution of NOCs. Using single-particle aerosol time-of-flight mass spectrometry, Wang et
al. (2010) observed that the widespread occurrence of NOCs closely correlated with particle
acidity in the atmosphere of Shanghai (China). In addition, real-time measurements of the
atmosphere in New York (US) by aerosol mass spectrometry indicated a definite link
between the age of organic species and the N/C ratio (Sun et al., 2011). Further in-depth
studies are required to identify the role of formation conditions (e.g., relative humidity (RH)
and pH) for secondary NOCs (Aiona et al., 2017; Nguyen et al., 2012). In the present study,
the mixing state of individual particles was investigated, involving NOCs, oxidized organics,
and ammonium, based on on-line seasonal observations using a single particle aerosol mass
spectrometry (SPAMS). Our findings show that the formation of NOCs is significantly
linked to oxidized organics and $NH_4^+$, which has important environmental implications for
assessing the impact and fate of these compounds.

**2 Methods**
**2.1 Field measurements**
Sampling was constructed at the Guangzhou Institute of Geochemistry, a representative
urban site in Guangzhou (China), a megacity in the Pearl River Delta (PRD) region. The size
and chemical composition of individual particles were obtained by the SPAMS (Hexin
Analytical Instrument Co., Ltd., China) in real-time (Li et al., 2011). The sampling inlet for
aerosol characterization was situated 40 meters above the ground level. A brief description
of the performance of the SPAMS and other instruments can be found in the Supporting
Information. The sampling periods covered four seasons, including spring (21/02 to 11/04
2014), summer (13/06 to 16/07 2013), autumn (26/09 to 19/10 2013), and winter (15/12 to
25/12 2013). The total measured particle numbers and mean values for meteorological data
and gaseous pollutants, are outlined for each season in Table S1 and were described in a
previous publication  (Zhang et al., 2019).

**2.2 SPAMS data analysis**

Fragments of NOCs were identified according to the detection of ion peaks at m/z -26

[CN]$^-$ or -42 [CNO]$^-$, generally due to the presence of C-N bonds (Silva and Prather, 2000;
Zawadowicz et al., 2017; Pagels et al., 2013). Laboratory produced C-N bonds compounds
from bulk solution-phase reactions between the representative oxidized organics (i.e.,
methylglyoxal) and ammonium sulfate was used to confirm the generation of ion peaks at
m/z -26 [CN]$^-$ and/or -42 [CNO]$^-$ using SPAMS (Fig. S1). Thus, the NOCs herein may refer
to complex nitrated organics such as organic nitrates, nitro-aromatics, nitrogen heterocycles,
and polyphenols. Unfortunately, how well [CN]$^-$/ [CNO]$^-$ ions could represent NOCs cannot
be quantified, although they were the most commonly reported NOCs peaks by single-
particle mass spectrometry (Silva and Prather, 2000; Zawadowicz et al., 2017; Pagels et al.,
2013). In the present study, [CN]$^-$/ [CNO]$^-$ ions are among the major peaks detected by the
SPAMS (Fig. 1). A rough estimate from the peak area ratio of [CN]$^-$/ [CNO]$^-$ ions and the
most likely NOCs fragments (i.e., various amines, and an entire series of nitrogen-containing
cluster ions $C_nN^-$, n = 1, 2, 3, …) (Silva and Prather, 2000) shows that [CN]$^-$/ [CNO]$^-$ ions
may represent more than 90% of these NOCs peaks. The number fractions (Nfs) of particles
that contained NOCs ranged from 56-59% across all four seasons (Table S1). The number
of detected NOCs-containing particles distributing along their vacuum aerodynamic
diameter ($d_{va}$) is shown in Fig. S2. Most of the detected NOC-containing particles had a $d_{va}$
in a range of 300-1200 nm.
A representative mass spectrum for NOCs-containing particles is shown in Fig. 1.
Dominant peaks in the mass spectrum were 39 $[K]^+$, 23 $[Na]^+$, nitrate (-62 $[NO_3]^-$ or -46
$[NO_2]^-$), sulfate (-97 $[HSO_4]^-$), organics (27 $[C_2H_3]^+$, 63 $[C_5H_3]^+$, -42 $[CNO]^-$, -26 $[CN]^-$),
ammonium (18 $[NH_4]^+$) and carbon ion clusters ($C_n^{+/-}$, n = 1, 2, 3,…). NOCs-containing
particles were internally mixed with various oxidized organics, represented as formate at m/z
-45 $[HCO_2]^-$, acetate at m/z -59 $[CH_3CO_2]^-$, methylglyoxal at m/z -71 $[C_3H_3O_2]^-$, glyoxylate
at m/z -73 $[C_2HO_3]^-$, pyruvate at m/z -87 $[C_3H_3O_3]^-$, malonate at m/z -103 $[C_3H_3O_4]^-$ and
succinate at m/z -117 $[C_4H_5O_4]^-$ (Zhang et al., 2017; Zauscher et al., 2013; Lee et al., 2003).
These oxidized organics showed their pronounced diurnal trends with afternoon maximum
and were highly correlated (r = 0.72 - 0.94, $p < 0.01$) with each other. Therefore, they were
primarily attributed to secondary oxidized organics from photochemical oxidation products
of various volatile organic compounds (VOCs) (Paulot et al., 2011; Zhao et al., 2012; Ho et
al., 2011), and the details can be found in our previous publication (Zhang et al., 2019). More
information on the seasonal variation range of the Nfs of oxidized organics, ammonium and
NOCs is presented in Fig. S3.
Hourly mean Nfs and relative peak areas were applied herein to indicate the variations
of aerosol compositions in individual particles. Even though advances have been made in
the quantification of specific chemical species for individual particles based on their
respective peak area information, it is still quite a challenge for SPAMS to provide
quantitative information on aerosol components mainly due to matrix effects, incomplete
ionization and so forth (Qin et al., 2006; Jeong et al., 2011; Healy et al., 2013; Zhou et al.,
2016). Despite this, the variation of relative peak area should be a good indicator for the
investigation of atmospheric processing of various species in individual particles (Wang et
al., 2010; Zauscher et al., 2013; Sullivan and Prather, 2007; Zhang et al., 2014).

**3 Results and Discussion**
**3.1 Evidence for the formation of NOCs from oxidized organics and ammonium**
Figure 2 shows the seasonal variations in Nfs of the oxidized organics and ammonium,
which were internally mixed with NOCs. On average, more than 90% of the oxidized
organics and 65% of ammonium (except spring) were found to be internally mixed with
NOCs (Fig. S4). Regarding that the Nfs of NOCs relative to all the measured particles was
~60%, it could be concluded that NOCs were enhanced with the presence of oxidized
organics and ammonium, with the enhancement associated with oxidized organics being the
most pronounced.
A strong correlation between both the Nfs and relative peak areas (RPAs) of NOCs and
oxidized organics further demonstrates their close associations, as shown in Fig. 3.
Compared with the oxidized organics, the Nfs of ammonium-containing particles internally
mixed with NOCs varied within a broader range (~40-90%). However, there is still an
enhancement mixing of NOCs with ammonium. A positive correlation ($R^2 = 0.50$, $p < 0.01$)
is observed between the hourly detected number of NOCs and ammonium. It is worth noting
that a negative correlation ($R^2 = 0.55$, $p < 0.01$) is obtained between the hourly average RPAs
of NOCs and ammonium (Fig. 3).

Based on both the enhancement of NOCs and the high correlations with oxidized

organics and ammonium, it is hypothesized that interactions between oxidized organics and
ammonium contributed to the observed NOCs. The formation of NOCs from ammonium
and carbonyls has been confirmed in several laboratory studies (Sareen et al., 2010; Shapiro
et al., 2009; Noziere et al., 2009; Kampf et al., 2016; Galloway et al., 2009). Secondary
organic aerosols (SOA) produced from a large group of biogenic and anthropogenic VOCs
can be further aged by $NH_3/NH_4^+$ to generate NOCs (Nguyen et al., 2012; Bones et al., 2010;
Updyke et al., 2012; Liu et al., 2015; Huang et al., 2017). In a chamber study, the formation
of NOCs is enhanced in an $NH_3$-rich environment (Chu et al., 2016). While such chemical
mechanisms might be complicated, the initial steps generally involve reactions forming
imines and amines, which can further react with carbonyl SOA compounds to form more
complex products (e.g., oligomers/BrC) (Laskin et al., 2015).

To verify this hypothesis, multiple linear regression analysis is performed to test how

well the RPAs of NOCs could be predicted by the oxidized organics and ammonium. As
expected, there is a close association ($R^2 = 0.71$, $p < 0.01$) between the predicted RPAs and
the observed values of NOCs (Fig. 4), which supports this hypothesis. A noticeable
improvement in $R^2$ implies that a model that uses both oxidized organics and ammonium to
predict RPAs of NOCs is substantially better than one that uses only one predictor (either
oxidized organics or ammonium in Fig. 3). The result indicates that interactions involving
oxidized organics and ammonium could explain over half of the observed variations in
NOCs in the atmosphere of Guangzhou. A fraction of the unaccounted NOCs could be due
to primary emissions and other formation pathways. This hypothesis could also be supported
by a similar pattern of diurnal variation observed for NOCs and oxidized organics (Fig. S5),
although there is a slight lag for the NOCs. Such a diurnal pattern is similar to those observed
in Beijing and Uintah (Yuan et al., 2016; Zhang et al., 2015). Notably, such a diurnal pattern
of secondary NOCs is adequately modelled when the production of NOCs via carbonyls and
ammonium is included (Woo et al., 2013). In addition to possible photo-bleaching (Zhao et
al., 2015), the lower contribution of NOCs during the daytime may be partly explained by
the lower RH, as discussed in section 3.2.

Interestingly, the relationship between NOCs and ammonium is distinctly different from

the relationship between NOCs and oxidized organics (Fig. 3). This implies that the
controlling factors on the formation of NOCs from ammonium are different from oxidized
organics. On the one hand, the positive correlation between the detected numbers reflects
that the formation of NOCs does require the participant of $NH_3/NH_4^+$, consistent with the
enhancement of NOCs in ammonium-containing particles (Fig. 2) discussed above. On the
other hand, the negative correlation between the RPAs signifies that the formation of NOCs
is most probably influenced by the relative amount of ammonium in individual particles.
Such influence could also be supported by our data, both from filter samples and individual
particle analysis. There is a negative correlation between concentrations of WSON and NH
$_4^+$ for the filter samples (Fig. S6). It can be seen from Fig. S7 that lower RPAs of ammonium

correspond to higher Nfs of ammonium that internally mixed with NOCs. Such an inverse

correlation could also serve as evidence to explain the influence of the relative amount of

ammonium on the formation of NOCs.

The influence of relative ammonium amount on the formation of NOCs is also

theoretically possible since the formation of NOCs may be enhanced by particle acidity

(Miyazaki et al., 2014; Aiona et al., 2017; Nguyen et al., 2012), which is substantially

affected by the abundance of ammonium. Consistently, higher relative acidity was observed

for the internally mixed ammonium and NOCs particles, compared to ammonium-containing

particles without NOCs (Fig. S6) and thus may influence the formation of NOCs (Fig. S7).

Particle acidity could also play a significant role in the gas-to-particle partitioning of

aldehydes (Herrmann et al., 2015; Liggio et al., 2005; Gen et al., 2018; De Haan et al., 2018;

Kroll et al., 2005), precursors for the formation of oxidized organics. However, the higher

relative acidity might also be a result of NOCs formation. A model simulation shows that

after including the chemistry of SOA ageing with $NH_3$, an increase in aerosol acidity would

be expected due to the reduction in ammonium (Zhu et al., 2018). It is also noted that the

particle acidity is roughly estimated by the relative abundance of ammonium, nitrate, and

sulfate in individual particles (Denkenberger et al., 2007), and thus may not be representative

of actual aerosol acidity or pH (Guo et al., 2015; Hennigan et al., 2015; Murphy et al., 2017).

In addition, ammonia in the gas phase is also efficient at producing NOCs (Nguyen et al.,

2012), which may play an intricate role in the distribution of ammonium and NOCs in the

particulate phase. The formation of ammonium and NOCs would compete for ammonia,

which may also potentially result in the negative correlation between the RPAs of NOCs and
ammonium. Unfortunately, such a role remains unclear since the variations of ammonia were
not available in the present study.

**3.2 Factors contributing to the NOCs resolved by positive matrix factorization (PMF)**
**analysis**

Figure 5 presents the PMF factor profiles obtained from the PMF model analysis

(detailed information is provided in the SI) (Norris et al., 2009) and their diurnal variations.
Around 75% of NOCs could be well explained by two factors, with 33% of the PMF resolved
NOCs mainly associated with ammonium and carbonaceous ion peaks (ammonium factor),
while 59% were mainly associated with oxidized organics (oxidized organics factor). The
explained fraction of NOCs by the ammonium and oxidized organic factors is consistent
with the linear regression analysis. Furthermore, PMF analysis provided information on the
factor contribution and diurnal variations, which may help explain the seasonal variations
and processes of NOCs. The ammonium factor showed a diurnal variation pattern peaking
during the early morning, which is consistent with the diurnal variation in RH (Zhang et al.,
2019). This factor contributed to ~80% (Fig. S8) of the PMF resolved NOCs during spring
with the highest RH (Table S1), whereas the oxidized organics factor dominated ($> 80\%$) in
summer and fall. In winter, these two factors similarly contributed (~40%). Variation of the
ammonium factor may reflect a potential role of aqueous pathways in the formation of NOCs,
particularly during spring. Differently, the oxidized organics factor showed a pattern of
diurnal variation, increasing from morning hours and peaking overnight, which may
correspond to the photochemical production of oxidized organics and followed interactions
with condensed ammonium. This pathway may explain the slightly late peaking of NOCs
compared to oxidized organics, as ammonium condensation is favorable overnight (Hu et al.,
2008). While there were similarities in the fractions of oxidized organics in the oxalate factor
and the oxidized organics factor, they only contributed to 8% of the PMF resolved NOCs in
the oxalate factor, which contained ~80% of the PMF resolved oxalate. As previously
discussed, these oxidized organics are also precursors for the formation of oxalate (Zhang et
al., 2019). Therefore, the PMF results suggest that there are two competitive pathways for
the evolution of these oxidized organics. Some oxidized organics formed from
photochemical activities were further oxidized to oxalate, resulting in a diurnal pattern of
variation with concentration peaks during the afternoon (Fig. 5), while others interact with
$NH_3/NH_4^+$ to form NOCs, peaking during the nighttime. However, the controlling factors for
these pathways could not be determined in the present study. The unexplained NOCs (~25%)
might be linked to the primary emissions, such as biomass burning (Desyaterik et al., 2013).
It could be partly supported by the presence of potassium and various carbon ion clusters (C
$_n^{+/-}$, n = 1, 2, 3, …) in the mass spectrum of NOCs-containing particles (Fig. 1).

**3.3 Seasonal variations in the observed NOCs**

There is an evident seasonal variation of NOCs, with higher relative contributions

during summer and autumn (Figs. 3 and 4), mainly due to the variations in oxidized organics
and $NH_3/NH_4^+$. In this region, a more considerable contribution from secondary oxidized
organics is typically observed during summer and autumn (Zhou et al., 2014; Yuan et al.,
2018). The seasonal maximum $NH_3$ concentrations have also been reported during the
warmer seasons, corresponding to the peak emissions from agricultural activities and high
temperatures, while the low $NH_3$ concentrations observed in colder seasons may be
attributed to gas-to-particle conversion (Pan et al., 2018; Zheng et al., 2012). Such seasonal
variation in NOCs is also obtained in a model simulation, showing that the conversion of
$NH_3$ into NOCs would result in a significantly higher reduction of gas-phase $NH_3$ during
summer (67%) than winter (31%), due to the higher $NH_3$ and SOA concentrations present in
the summer (Zhu et al., 2018). More primary NOCs may also be present during summer and
autumn in the present study, due to the additional biomass burning activities in these seasons
(Chen et al., 2018; Zhang et al., 2013).
The seasonal variations of NOCs can be adequately explained by the variations in
concentrations of oxidized organics and ammonium (Fig. 4), although the hourly variations
during each season are not well explained, as indicated by the lower $R^2$ values (Table S2).
The correlation coefficients ($R^2$) range from 0.24 to 0.57 for inter-seasonal variations.
During spring, NOCs exhibits a limited dependence on oxidized organics (Figs. 3a and 3b),
while during summer, the hourly detected number of NOCs shows a limited dependence on
ammonium (Fig. 3d). These seasonal dependences of NOCs are consistent with the PMF
results, showing that the ammonium factor explained ~80% of the predicted NOCs during
spring, while the oxidized organics factor dominantly contributed to the predicted NOCs
during warmer seasons (Fig. S8). A detailed discussion of this issue is provided in the SI.

**3.4 Influence of RH and NOx**
The influence of RH on RPAs of NOCs and peak ratios of NOCs/oxidized organics are
shown in Fig. 6. While NOCs do not show a clear dependence on RH, the ratio of NOCs to
the oxidized organics shows an apparent increase towards higher RH. This finding is
consistent with the observations reported by Xu et al. (2017), in which the N/C ratio
significantly increases as a function of RH in the atmosphere of Beijing. Besides, the diurnal
variations of NOCs with peaks values around 20:00 are also similar to those reported by Xu
et al. (2017). The peak ratios of NOCs/oxidized organics are more obviously enhanced when
RH is higher than 40%. These findings imply that aqueous-phase processing likely plays a
substantial role in the formation of NOCs. Significant changes in RH, such as during the
evaporation of water droplets, have been reported to facilitate the formation of NOCs via
$NH_3/NH_4^+$ and SOA (Nguyen et al., 2012). In addition, an increase in RH would improve
the uptake of $NH_3$ and the formation of $NH_4^+$, which also contributes to the enhancement of
NOCs. However, the relatively weak correlation ($R^2 = 0.27$, $p < 0.01$) between the peak
ratios and RH, reflect the complex influence of RH on the formation of NOCs (Xu et al.,
2017; Woo et al., 2013).
One may expect that NOCs are formed through the interactions between NOx and
oxidized organics in the gas phase, followed by condensation (Fry et al., 2014; Lehtipalo et
al., 2018). Similar to that observed for RH, while NOCs do not show a clear dependence on
NOx (Fig. 6c, $R^2$ = 0.02–0.13), the ratio of NOCs to the oxidized organics shows a clear
increasing trend towards higher NOx (Fig. 6d, $R^2$ = 0.18, $p < 0.01$). This indicates that NOx
may play a certain role in the conversion of oxidized organics to NOCs, and yet it cannot be
quantified in the present study. It is also noted that low correlation coefficients between NOx
and NOCs might not indicate a limited contribution of NOx to the formation of NOCs. NOx
affects the formation of NOCs in various ways (e.g., peroxy radical chemistry in VOCs
oxidation mechanisms and formation of nitrate radicals) {Xu, 2015 #20234}{Zhang, 2018
#22855}, and thus may not linearly contribute to the formation of NOCs.

**3.5 Atmospheric implications and limitation**

In this study, we showed that in an urban megacity area, secondary NOCs were

significantly contributed by the heterogeneous ageing of oxidized organics with $NH_3/NH_4^+$,
providing valuable insight into SOA aging mechanisms. In particular, the effects of $NH_3/NH_4^+$
on SOA or BrC formation remain relatively poorly understood. In the PRD region, it has
been shown that oxygenated organic aerosols (OOA) account for more than 40% of the total
organic mass (He et al., 2011), with high concentrations of available gaseous carbonyls (Li
et al., 2014). Therefore, it is expected that over half of all water-soluble NOCs in this region
might link to secondary processing (Yu et al., 2017). Furthermore, secondary sources have
been found to contribute significantly to NOCs related BrC in Nanjing, China (Chen et al.,
2018). The results presented herein also suggest that the production of NOCs might be
adequately estimated by their correlation with secondary oxidized organics and ammonium.
The effectiveness of correlation-based estimations needs to be examined in other regions
before being generally applied in other environments. However, this approach may provide
valuable insights into investigations of NOCs using atmospheric observations. In contrast, it
has previously been reported that a positive correlation exists between WSON and
ammonium (Li et al., 2012), indicating similar anthropogenic sources. This divergence could
be mainly attributed to varying contributions of primary sources and secondary processes to
the observed NOCs. Possible future reductions in anthropogenic emissions of ammonia may
reduce particle NOCs. Understanding the complex interplay between inorganic and organic
nitrogen is an essential part of assessing global nitrogen cycling.

Moise et al. (2015) proposed that with high concentrations of reduced nitrogen

compounds, high photochemical activity, and frequent changes in humidity, BrC formed via
$NH_3/NH_4^+$ and SOA may become a dominant contributor to aerosol absorption, specifically
in agricultural and forested areas. However, this study suggests that even in typical urban
areas, BrC formation via $NH_3/NH_4^+$ and SOA should not be neglected. In particular, SOA
was found to account for 44 – 71% of the organic mass in megacities across China (Huang
et al., 2014), with $NH_3$ concentrations in urban areas comparable with those from agricultural
sites and 2- or 3-fold those of forested areas in China (Pan et al., 2018). Additionally, the
acidic nature of particles in these regions would also be favorable for the formation of NOCs
(Guo et al., 2017; Jia et al., 2018). Considering the formation of NOCs from the uptake of
NH$_3$ onto SOA particles, Zhu et al. (2018) suggested that this mechanism could have a
significant impact on the atmospheric concentrations of NH$_3$/NH$_4^+$ and NO$_3^-$.

**5 Conclusions**

This study investigated the processes contributing to the seasonal formation of NOCs,

involving ammonium and oxidized organics in urban Guangzhou, using single-particle mass
spectrometry. This is the first study to provide direct field observation results to confirm that
the variation of NOCs correlate well and are strongly enhanced internal mixing with
secondary oxidized organics. These findings highlight the possible formation pathway of
NOCs through the ageing of secondary oxidized organics by NH$_3$/NH$_4^+$ in ambient urban
environments. A clear pattern of seasonal variation in NOCs was observed, with higher
relative contributions in summer and autumn as compared to spring and winter. This
seasonal variation was well predicted by multiple linear regression model analysis, using the
relative abundance of oxidized organics and ammonium as model inputs. More than 50% of
NOCs could be explained by the interaction between oxidized organics and ammonium. The
production of NOCs through such processes was facilitated by increased humidity and NOx.
These results extend our understanding of the mixing state and atmospheric processing of
particulate NOCs, as well as having substantial implications for the accuracy of models
predicting the formation, fate, and impacts of NOCs in the atmosphere.

**Author contribution**
GHZ and XHB designed the research (with input from WS, LL, ZYW, DHC, MJT, XMW
and GYS), analyzed the data, and wrote the manuscript. XFL, YZF, and QHL conducted air
sampling work and laboratory experiments under the guidance of GHZ, XHB and XMW.
All authors contributed to the refinement of the submitted manuscript.

**Acknowledgement**
This work was supported by the National Nature Science Foundation of China (No.
41775124 and 41877307), the National Key Research and Development Program of China
(2017YFC0210104 and 2016YFC0202204), the Science and Technology Project of
Guangzhou, China (No. 201803030032), and the Guangdong Foundation for Program of
Science and Technology Research (No. 2017B030314057).

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

**Figure captions**

Figure 1. Representative mass spectrum for NOCs-containing particles. The ion

peaks corresponding to NOCs and oxidized organics are highlighted with red bars.

Figure 2. The variation in hourly mean Nfs of the oxidized organics and

ammonium that internally mixed with NOCs. Box and whisker plot shows lower,
median, and upper lines, denoting the $25^{th}$, $50^{th}$, and $75^{th}$ percentiles, respectively; the
lower and upper edges denote the $10^{th}$ and $90^{th}$ percentiles, respectively.

Figure 3. Correlation analysis of (a, c) the RPAs and (b, d) the number of

detected NOCs, with the oxidized organics and ammonium in different seasons.
Significant ($p < 0.01$) correlations were obtained for both the total observed data and
the seasonally separated data. RPA is defined as the fractional peak area of each m/z
relative to the sum of peak areas in the mass spectrum and is applied to represent the
relative amount of a species on a particle (Jeong et al., 2011; Healy et al., 2013).

Figure 4. Comparison between the measured and predicted RPAs for NOCs.

Figure 5. (left) PMF-resolved 3-factor source profiles (percentage of total species)

and (right) their diurnal variations (arbitrary unit).

Figure 6. The dependence of NOCs and the ratio of NOCs to the oxidized organics

on RH.

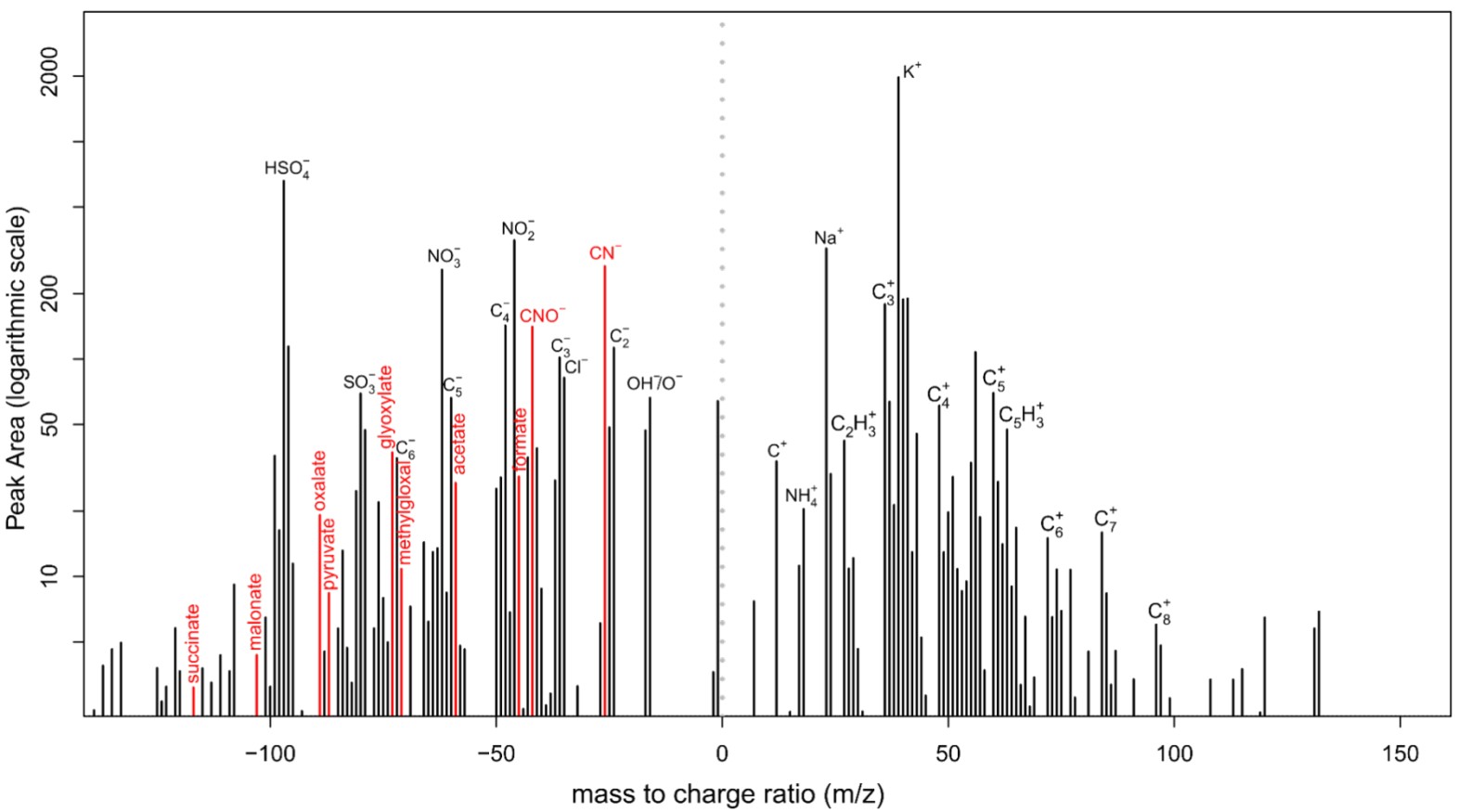


Fig. 1.

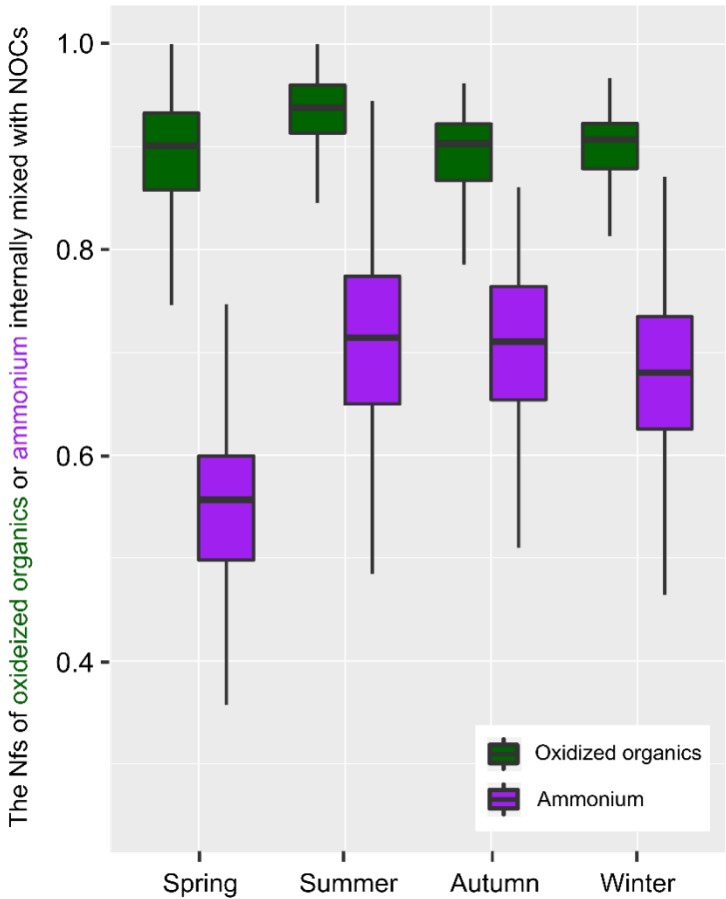


Fig. 2.

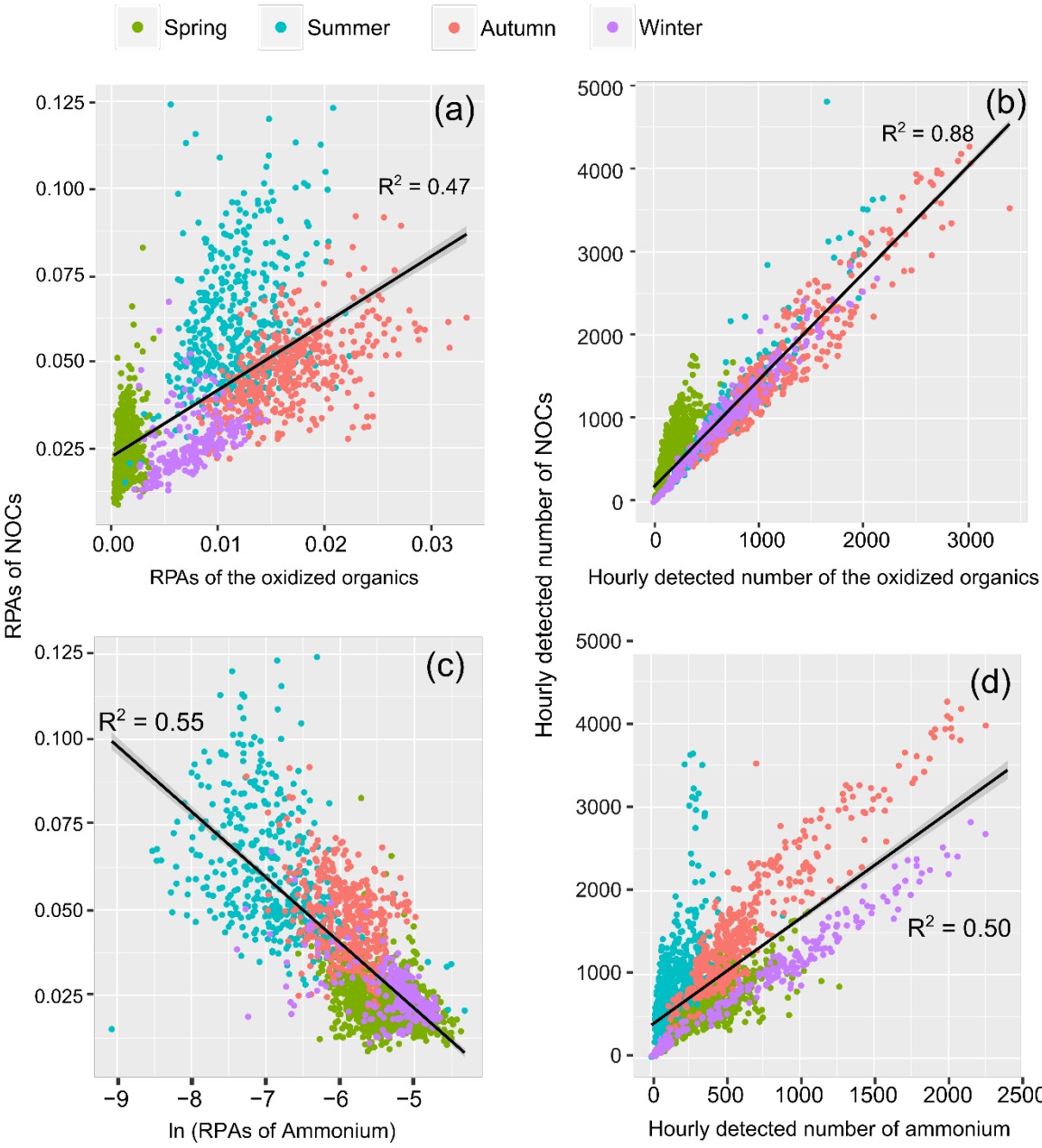


Fig. 3.

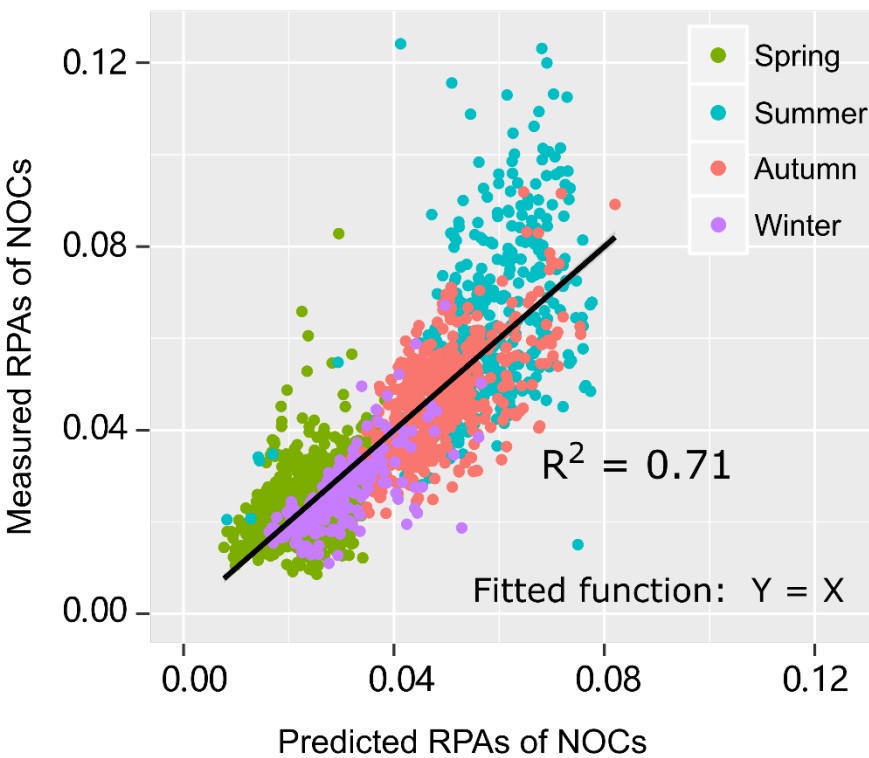


**Fig. 4.**


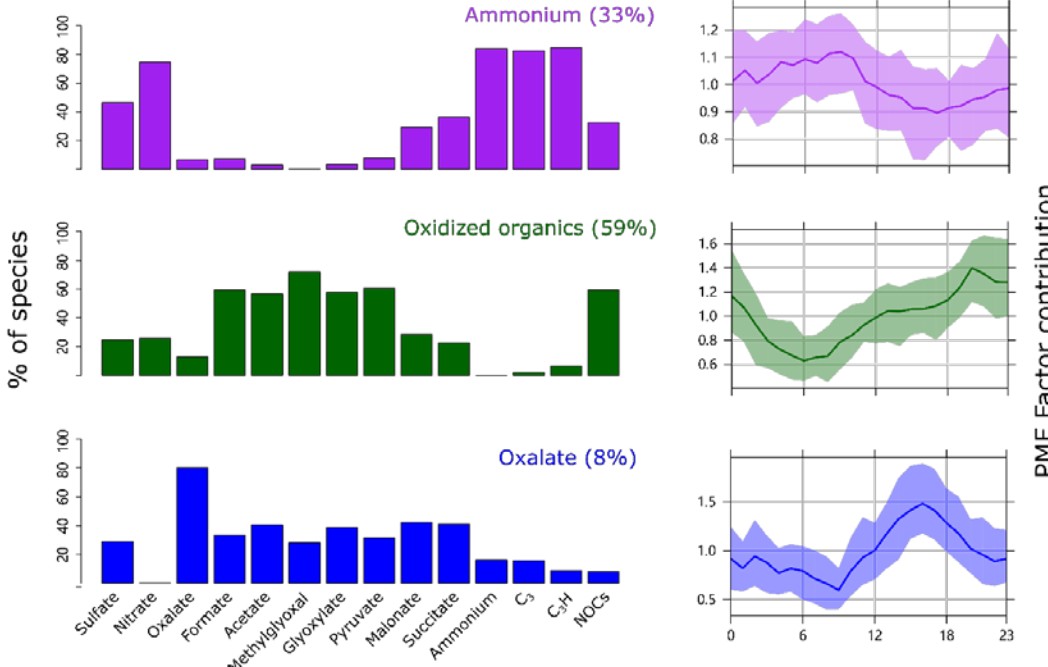


**Fig. 5.**

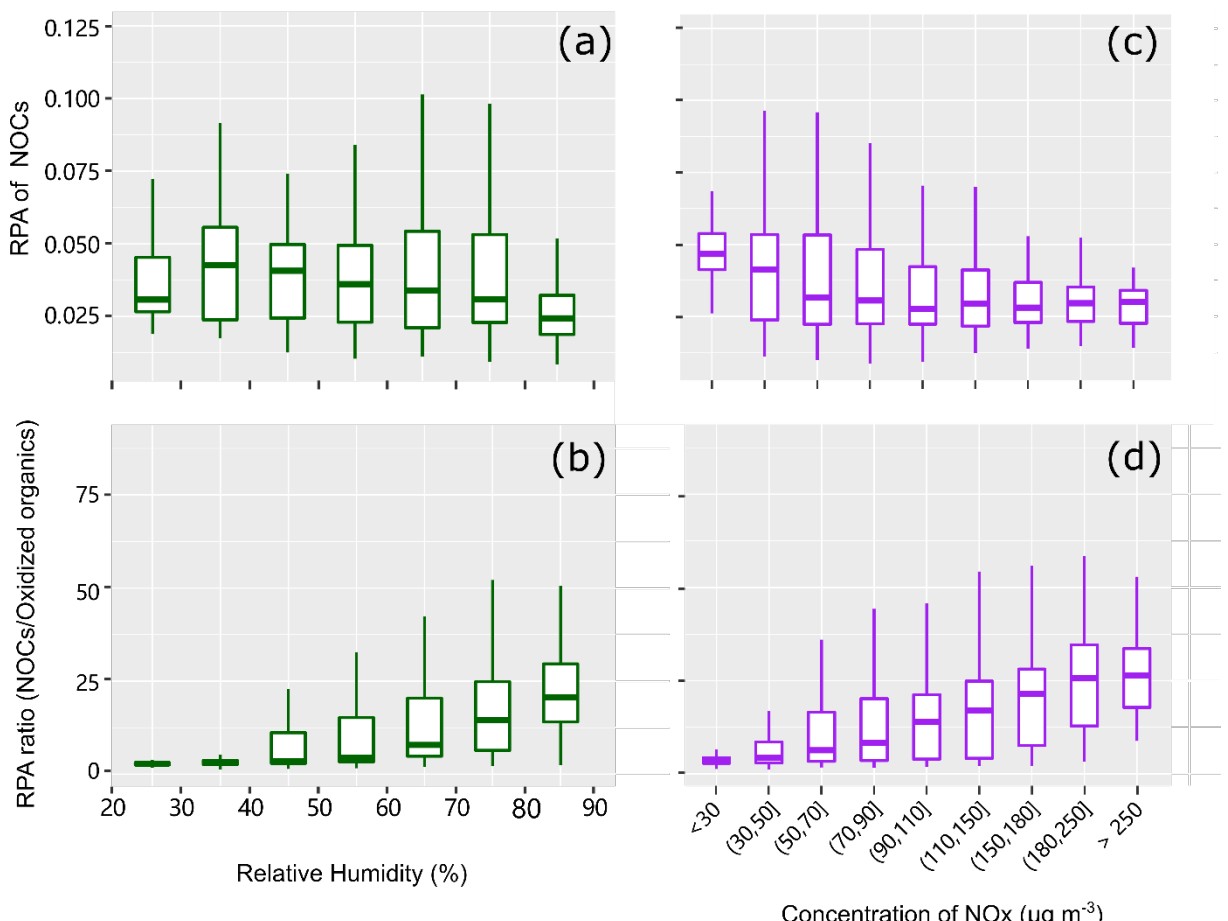


**Fig. 6.**