# Peer review of "High secondary formation of nitrogen-containing organics (NOCs) and its possible link to oxidized organics and ammonium"

_Atmospheric Chemistry and Physics, 2019_

## Referee Comment (RC1) · Anonymous Referee #1 · 20 Aug 2019

Review of "High secondary formation of nitrogen-containing organics (NOCs) and its possible link to oxidized organics and ammonium" by G. Zhang et al.

**General Comments:**
This study presents the results of single-particle measurements in Guangzhou, China. The focus is on nitrogen-containing organic compounds (NOC) and the role of ammonium in NOC formation. The highly time resolved measurements span four seasons, and thus could offer new insight into NOC chemistry. While the data set is unique and the topic is certainly appropriate for *Atmospheric Chemistry and Physics*, there are quite a few major issues with the study – including analysis methods, assumptions, data interpretations, and conclusions – that prevent me from recommending it for publication at this time. It is possible that these issues could be addressed with a major revision, but not guaranteed. My specific concerns are addressed below.

**Specific Comments:**

1. The authors come to the conclusion that NOC formation during their study is not likely from gas-phase reactions, but is predominantly from heterogeneous/particle-phase reactions. Their logic for this argument is quite confusing. I do not believe this conclusion is at all supported by the data presented in the manuscript.

2. Similarly, I think the explanations for the role of $NO_x$ and $NH_3$ in particle NOC formation are extremely muddled. For $NH_3$, a positive correlation is observed between the number fraction of particles with $NH_4^+$ and NOC, while a negative correlation is observed between the relative peak areas of these compound classes. There is not a reasonable explanation given for this surprising and apparently contradictory behavior. Further, the manuscript mostly discounts the negative correlation in the relative peak areas, instead assuming that $NH_3$ drives (or is prominently involved in) NOC formation. $NO_x$ is completely ruled out as a contributor to NOC formation on the basis of poor (or no) correlations between NOC and $NO_x$. However, this is a misinterpretation of the data. Many factors (different removal processes and lifetimes of particles vs. gasses, primary vs. secondary species, etc.) could contribute to a lack of correlation even if $NO_x$ did contribute to NOC formation. As the data are currently presented and explained, it is completely unclear how $NH_3$ or $NO_x$ play a role in NOC formation in the present study, although such analyses should be possible with their data set.

3. The assumptions and discussion related to particle acidity (lines 307 – 337, Figure 7) are not correct. Recent studies have shown that ratios of aerosol inorganics (including variations involving NH4-SO4-NO3) are not suitable proxies for particle acidity (Guo et al., 2015; Hennigan et al., 2015; Murphy et al., 2017). Also, the discussion of acid-catalyzed SOA (lines 375-381) is not correct, so the implications of the present study are misstated.

4. Finally, the application, interpretation, and discussion of PMF and multiple linear regression methods need substantial revision. The explanation of the PMF approach is quite confusing, and as it is written, does not add anything substantive beyond the general correlations presented before it. The multiple linear regression also does not support any of the stated conclusions, beyond what was already presented for the individual correlations to $NH_4^+$ and

oxygenated organics. The discussion of "modeleld NOCs" (e.g., lines 291, 374, Fig. 4) is misleading, especially compared to how this is typically used in atmospheric studies.

**Technical Corrections:**
The above issues are substantial enough that any technical corrections can be addressed on review of the revised manuscript.

**References:**

Guo, H., Xu, L., Bougiatioti, A., Cerully, K. M., Capps, S. L., Hite Jr., J. R., Carlton, A. G., Lee, S.-H., Bergin, M. H., Ng, N. L., Nenes, A., and Weber, R. J.: Fine-particle water and pH in the southeastern United States, Atmos. Chem. Phys., 15, 5211-5228, https://doi.org/10.5194/acp-15-5211-2015, 2015.

Hennigan, C. J., Izumi, J., Sullivan, A. P., Weber, R. J., and Nenes, A.: A critical evaluation of proxy methods used to estimate the acidity of atmospheric particles, Atmos. Chem. Phys., 15, 2775–2790, https://doi.org/10.5194/acp-15-2775-2015, 2015.

Murphy, J. G., Gregoire, P. K., Tevlin, A. G., Wentworth, G. R., Ellis, R. A., Markovic, M. Z., and VandenBoer, T. C.: Observational constraints on particle acidity using measurements and modelling of particles and gases, Faraday Discuss., 200, 379-395, https://doi.org/10.1039/c7fd00086c, 2017.

---

## Referee Comment (RC2) · Anonymous Referee #2 · 21 Aug 2019

This paper analyzed single particle aerosol mass spectrometer (SPAMS) data for ambient aerosols and found there are relations between CN-/CNO- ion intensities and some other species, such as oxidized organic ions and ammonium. It is an interesting report. But there are some concerns which need to be addressed before publication.

One of the major problems is that this paper attribute oxidized organics to secondary formation. However, it may not be the case. Biomass burning or coal combustion can also produce oxidized organics including large amounts of NOCs. Actually, in many previous single particle mass spectrometry studies, CN- and CNO- were taken as ion markers for combustion sources. The authors need to provide more evidences either

to rule out the possibility of primary oxygenated organics and primary NOCs or to distinguish the secondary organics from the primary ones.

Another major concern is that how well CN-/CNO- ions can represent total NOCs. Can they represent 25%, 50% or 75% of total NOCs? The paper needs to provide more discussion on this issue.

The third concern is that ammonium sulfate is very difficult to be ionized under 266 nm UV laser. Thus, it is likely that some mass spectra of particles do not contain NH4+ peak but these particles may still contain ammonium sulfate. The authors also need to provide some discussions on this possibility.

Specific comments:

Line 54: how much is "large"? It would be always better to provide a number or range.

Line 149: "so on" is a bit informal. I would change "so on" to "so forth"

Line 220: How do you come up with this statement: "...explain over half of the observed variations in NOCs in the atmosphere of Guangzhou."? Please elaborate and provide more details.

Line 224: Please report if the PMF analysis reaches convergence or not. How much is the error of the PMF modelling in the paper?

Line 387: Check English

---

## Author Comment (AC1) · 24 Oct 2019

**Response to comments**

Review of "High secondary formation of nitrogen-containing organics (NOCs) and its possible link to oxidized organics and ammonium" by G. Zhang et al.

**General Comments:**

This study presents the results of single-particle measurements in Guangzhou, China. The focus is on nitrogen-containing organic compounds (NOC) and the role of ammonium in NOC formation. The highly time resolved measurements span four seasons, and thus could offer new insight into NOC chemistry. While the data set is unique and the topic is certainly appropriate for Atmospheric Chemistry and Physics, there are quite a few major issues with the study – including analysis methods, assumptions, data interpretations, and conclusions – that prevent me from recommending it for publication at this time. It is possible that these issues could be addressed with a major revision, but not guaranteed. My specific concerns are addressed below.

We would like to thank the reviewer for his/her useful comments and recommendations to improve the manuscript. We have addressed the specific comments in the sections below and made the appropriate revisions to the manuscript. Reviewer comments are in black text followed by our response in blue text.

**Specific Comments:**

1. The authors come to the conclusion that NOC formation during their study is not likely from gas-phase reactions, but is predominantly from heterogeneous/particle-phase reactions. Their logic for this argument is quite confusing. I do not believe this conclusion is at all supported by the data presented in the manuscript.

Thanks for the constructive comment. The reviewer thought our hypothesis that NOCs formation from oxidized organics and ammonium was confusing, which is largely because the supporting evidence was not separately discussed and emphasized. In the revised manuscript, therefore, we reorganized the text and emphasized the evidence to make the argument more clear. Based on both the enhancement of NOCs and the high correlations with oxidized organics and ammonium, a hypothesis is put forward: interactions between oxidized organics and ammonium contribute to the observed NOCs. To support such hypothesis, further evidence such as diurnal variations of NOCs, filter measurements and multiple linear regression analysis are provided and discussed. Supporting evidence from previous laboratory and modeling studies is also discussed. Please refer to section 3.1 for more details.

2. Similarly, I think the explanations for the role of NOx and NH3 in particle NOC formation are extremely muddled. For NH3, a positive correlation is observed between the number fraction of particles with NH4 + and NOC, while a negative correlation is observed between the relative peak areas of these compound classes. There is not a reasonable explanation given for this surprising and apparently contradictory behavior. Further, the manuscript mostly discounts the negative correlation in the relative peak areas, instead assuming that NH3 drives (or is prominently involved in) NOC formation. NOx is completely ruled out as a contributor to NOC formation on the basis of poor (or no) correlations between NOC and NOx. However, this is a misinterpretation of the data. Many factors (different removal processes and lifetimes of particles vs. gasses, primary vs. secondary species, etc.) could contribute to a lack of correlation even if NOx did contribute to NOC formation. As the data are currently presented and explained, it is completely unclear how NH3 or NOx play a role in NOC formation in the present study, although such analyses should be possible with their data set.

Thanks for the comment. We also think that it is a surprising and interesting results for the contradictory correlation between NOCs and ammonium. We have highlighted it as "Interestingly, the relationship between NOCs and ammonium was distinctly different from the relationship between NOCs and oxidized organics." and "A positive correlation ($R^2 = 0.50$, $p < 0.01$) is observed between the hourly detected number of NOCs and ammonium. It is worth noting that a negative correlation ($R^2 = 0.55$, $p < 0.01$) is obtained between the hourly average RPAs of NOCs and ammonium (Fig. 3)." in Lines 176-179 and 201-202.

To make the statement clearer, we highlighted the reasoning in two parts in the revised manuscript as "Interestingly, the relationship between NOCs and ammonium is distinctly different from the relationship between NOCs and oxidized organics (Fig. 3). This implies that the controlling factors on the formation of NOCs from ammonium are different from oxidized organics. On one hand, the positive correlation between the detected numbers reflects that the formation of NOCs does require the participant of $NH_3/NH_4^+$ , consistent with the enhancement of NOCs in ammonium-containing particles discussed above. On the other hand, the negative correlation between the RPAs signifies that particles with higher relative ammonium content may inhibit the formation of NOCs.". Please refer to Lines 209–216.

For the role of NOx on the formation of NOCs, we agree with the comment that many factors (different removal processes and lifetimes of particles vs. gasses, primary vs. secondary species, etc.) could contribute to a lack of correlation even if NOx did contribute to NOC formation. In the Lines 319-328 of the revised manuscript, we have added this probability in our discussion, as "Low correlation coefficients ($R^2 = 0.02$–0.13) between NOCs and NOx likely indicates limited contribution of this pathways to the observed NOCs. We have also included an analysis on the relationship between peak ratios of NOCs/oxidized organics and NOx. Peak area ratios of NOCs/oxidized organics generally increases with increasing level of NOx (Fig. 6), but still with relatively weak correlation ($R^2 = 0.18$, $p < 0.01$). An inclusion of both NOx and RH in the above linear regression model (NOCs versus the oxidized organics and ammonium) does not improve the prediction of NOCs ($R^2 = 0.71$, $p < 0.01$). However, it is also noted that many factors (e.g., different removal processes and lifetimes of particles vs. gasses, primary vs. secondary species, etc.) could contribute to a lack of strong correlation even if NOx did contribute to NOC formation.".

3. The assumptions and discussion related to particle acidity (lines 307 – 337, Figure 7) are not correct. Recent studies have shown that ratios of aerosol inorganics (including variations involving NH4-SO4-NO3) are not suitable proxies for particle acidity (Guo et al., 2015; Hennigan et al., 2015; Murphy et al., 2017). Also, the discussion of acid-catalyzed SOA (lines 375-381) is not correct, so the implications of the present study are misstated.

Thanks for pointing out the deficiency. We agree with the comment that ratios of aerosol inorganics are not suitable proxies for particle acidity. Therefore, we have shorten our discussion on this issue, and included these references to show that the estimated particle acidity may not be representative of actual aerosol acidity or pH. In the revised manuscript, we only intend to show that the variation of ammonium may affect the particle acidity in the following sentence: "…particles with higher relative ammonium content may inhibit the formation of NOCs. This is supported by the inverse correlation between that Nfs of ammonium that internally mixed with NOCs and the RPAs of ammonium (Fig. S7). This is also theoretically possible since the formation of NOCs may be influenced by particle acidity (Miyazaki et al., 2014; Aiona et al., 2017; Nguyen et al., 2012), which is substantially affected by the abundance of ammonium. Particle acidity could also play a significant role in the gas-to-particle partitioning of aldehydes (Herrmann et al., 2015; Liggio et al., 2005; Gen et al., 2018; De Haan et al., 2018; Kroll et al., 2005), precursors for the formation of oxidized organics. Consistently, higher relative acidity was observed for the internally mixed ammonium and NOCs particles, compared to ammonium-containing particles without NOCs (Fig. S6), and thus may influence the formation of NOCs (Fig. S7). However, the higher relative acidity might also be a result of NOCs formation. A model simulation shows that after including the chemistry of SOA ageing with $NH_3$, an increase in aerosol acidity would be expected due to the reduction in ammonium (Zhu et al., 2018). It is also noted that the particle acidity is roughly estimated by the relative abundance of ammonium, nitrate, and sulfate in individual particles (Denkenberger et al., 2007), and thus may not be representative of actual aerosol acidity or pH (Guo et al., 2015; Hennigan et al., 2015; Murphy et al., 2017).", please refer to Line 217-233.

4. Finally, the application, interpretation, and discussion of PMF and multiple linear regression methods need substantial revision. The explanation of the PMF approach is quite confusing, and as it is written, does not add anything substantive beyond the general correlations presented before it. The multiple linear regression also does not support any of the stated conclusions, beyond what was already presented for the individual correlations to NH4 + and oxygenated organics. The discussion of "modelled NOCs" (e.g., lines 291, 374, Fig. 4) is misleading, especially compared to how this is typically used in atmospheric studies.

Thanks for the comment. We think the reviewer's confusion about the explanation of PMF and linear regression analysis is largely because the additional information provided by PMF and linear regression analysis was not emphasized in the original manuscript. In the revised manuscript (section 3.2), we have highlighted the additional information provided by the PMF approach in a new section 3.2: Three PMF factors were resolved to explain the formation of NOCs. Around 75% of NOCs could be well explained by two factors, with 33% of the PMF resolved NOCs mainly associated with ammonium and carbonaceous ion peaks (ammonium factor), while 59% were mainly associated with oxidized organics (oxidized organics factor). The explained fraction of NOCs by the ammonium and oxidized organic factors is consistent with the linear regression analysis. In addition, PMF analysis provided information on the factor contribution and diurnal variations, which may help explain the seasonal variations and processes of NOCs. Profiles and their diurnal variations of these factors (Fig. 5) suggest that there were two competitive pathways for the evolution of these oxidized organics. Some oxidized organics formed from photochemical activities were further oxidized to oxalate, resulting in a diurnal pattern of variation with concentration peaks during the afternoon (Fig. 5), while others interact with $NH_3/NH_4^+$ to form NOCs, peaking during the nighttime.

The multiple linear regression applied herein is to show how well the observed NOCs could be reconstructed by the observed oxidized organics and ammonium. As expected, there is a close association ($R^2 = 0.71$, $p < 0.01$) between the predicted RPAs and the observed values of NOCs (Fig. 4), which supports this hypothesis. An obvious improvement in $R^2$ implies that a model that uses both oxidized organics and ammonium to predict RPAs of NOCs is substantially better than one that uses only one predictor (either oxidized organics or ammonium in Fig. 3). We understand that the use of "modelled NOCs" for the regression analysis may be misleading. In the revised manuscript, we revised the "modelled NOCs" to "PMF resolved NOCs" to avoid the misunderstanding.

**Technical Corrections:**

The above issues are substantial enough that any technical corrections can be addressed on review of the revised manuscript.

Thanks for the comment. We have carefully examined possible technical errors, including those raised by Refree 2#.

**References:**

[revised manuscript text omitted]
 a their close associations between these factors, as shown in Fig. 3. Compared with the variation in oxidized organics, the Nfs of ammonium- containing particles internally mixed with NOCs varied within a wider range (~40-90%).

However, there was is still an enhancement mixing of NOCs with ammonium. In addition, aA positive correlation ($R^2 = 0.50$, $p < 0.01$) is observed between the hourly detected number of NOCs and ammonium. In contrast,It is worth noting that a negative correlation ($R^2 = 0.55$,

$p < 0.01$) is obtained between the hourly average RPAs of NOCs and ammonium (Fig. 3).

Interestingly, the relationship between NOCs and ammonium was distinctly different from the relationship between NOCs and oxidized organics. A positive correlation ($R^2 = 0.50$, $p$

$< 0.01$) was observed between the hourly detected number of NOCs and ammonium. In contrast, a negative correlation ($R^2 = 0.55$, $p < 0.01$) was observed between the hourly average relative peak areas (RPAs) of NOCs and ammonium (Fig. 3).

Based on both the enhancement of NOCs and the high correlations with oxidized organics and ammonium, it is hypothesized that interactions between oxidized organics and ammonium contributed to the observed NOCsthe dominant association between oxidized organics and NOCs (Fig. 2) indicates that NOCs may be formed from the processing of secondary oxidized organics in particle phase, rather than gas phase reactions followed by condensation. Actually, formation of NOCs from ammonium and carbonyls has been confirmed in several laboratory studies (Sareen et al., 2010; Shapiro et al., 2009; Noziere et al., 2009; Kampf et al., 2016; Galloway et al., 2009). Secondary organic aerosols (SOA) produced from a large group of biogenic and anthropogenic VOCs can be further aged by $NH_3/NH_4^+$ to generate NOCs (Nguyen et al., 2012; Bones et al., 2010; Updyke et al., 2012; Liu et al., 2015; Huang et al., 2017). In a chamber study, the formation of NOCs were found to beis enhanced in a $NH_3$-rich environment (Chu et al., 2016). While such chemical mechanisms might be complex, the initial steps generally involve reactions forming imines and amines, which can further react with carbonyl SOA compounds to form more complex products (e.g., oligomers/BrC) (Laskin et al., 2015).

To verify this hypothesis, multiple linear regression analysis was is performed to test how well could the RPAs of NOCs could be predicted by the oxidized organics and ammonium. As expected, there is a close association ($R^2 = 0.71$, $p < 0.01$) between the predicted RPAs and the observed values of NOCs (Fig. 4), which supports this hypothesis. An obvious substantial improvement in $R^2$ implies that a model that uses both oxidized organics and ammonium to predict RPAs of NOCs is substantially better than one that uses only one predictor (either oxidized organics or ammonium in Fig. 3). The result indicates that interactions involving oxidized organics and ammonium could explain over half of the observed variations in NOCs in the atmosphere of Guangzhou. A fraction of the unaccounted NOCs could be due to primary emissions and other formation pathways.

Actually, formation of NOCs from ammonium and carbonyls have been confirmed in several laboratory studies (Sareen et al., 2010; Shapiro et al., 2009; Noziere et al., 2009;

Theis hypothesis could also be supported by the similar pattern of diurnal variation observed for NOCs and oxidized organics (Fig. S5), although there is a slight lag for the  NOCs. Such diurnal pattern is similar to those observed in Beijing and Uintah (Yuan et al., 2016; Zhang et al., 2015). Notably, such diurnal pattern of secondary NOCs is effectively modelled when the production of NOCs via carbonyls and ammonium is included (Woo et al., 2013). In addition to possible photo- bleaching (Zhao et al., 2015), the lower contribution of NOCs during daytime may be partly explained by the lower RH, as discussed in section 3.2.

Interestingly, the relationship between NOCs and ammonium is distinctly different from the relationship between NOCs and oxidized organics (Fig. 3).

a forest in northern Japan (Miyazaki et al., 2014). This is further supported by the similar pattern of diurnal variation observed for NOCs and oxidized organics (Fig. S6). However, a slight lag period was observed in the overnight peaks of NOCs, as compared to those of the oxidized organics. This finding was consistent with previously reported results, showing

NOCs to have concentration maxima overnight in Beijing and Uintah (Yuan et al., 2016;

Zhang et al., 2015). The lower contribution of NOCs during daytime may be partly explained by the lower RH, as discussed in section 3.2, in addition to photo-bleaching which occurs during daytime (Zhao et al., 2015).

Interestingly, the relationship between NOCs and ammonium was distinctly different from the relationship between NOCs and oxidized organics. A positive correlation ($R^2 =$

0.50, $p < 0.01$) was observed between the hourly detected number of NOCs and ammonium.

In contrast, a negative correlation ($R^2 = 0.55$, $p < 0.01$) was observed between the hourly average relative peak areas (RPAs) of NOCs and ammonium (Fig. 3). This implies that the controlling factors on the formation of NOCs from ammonium are different from those controlling oxidized organics. On one hand, the positive correlation between the detected numbers reflects that the formation of NOCs does require the participant of $NH_3/NH_4^+$, consistent with the enhancement of NOCs in ammonium-containing particles discussed above. On the other hand, the negative correlation between the RPAs signifies that particles with higher relative ammonium content may inhibit the formation of NOCs. the relative amount of ammonium may influences the formation of NOCs. Consistently, there is a negative correlation between concentrations of WSON and $NH_4^+$ in filter samples (Fig. S6).

This is supported by the inverse correlation between that Nfs of ammonium that internally mixed with NOCs and the RPAs of ammonium (Fig. S7). This is also theoretically possible since the formation of NOCs may be influenced by particle acidity (Miyazaki et al., 2014;

Aiona et al., 2017; Nguyen et al., 2012), which is substantially affected by the abundance of ammonium. Particle acidity could also play a significant role in the gas-to-particle partitioning of aldehydes (Herrmann et al., 2015; Liggio et al., 2005; Gen et al., 2018; De

Haan et al., 2018; Kroll et al., 2005), precursors for the formation of oxidized organics.

Consistently, higher relative acidity was observed for the internally mixed ammonium and

NOCs particles, compared to ammonium-containing particles without NOCs (Fig. S6), and thus may influence the formation of NOCs (Fig. S7). However, the higher relative acidity might also be a result of NOCs formation. A model simulation shows that after including the chemistry of SOA ageing with $NH_3$, an increase in aerosol acidity would be expected due to the reduction in ammonium (Zhu et al., 2018). It is also noted that the particle acidity is roughly estimated by the relative abundance of ammonium, nitrate, and sulfate in individual particles (Denkenberger et al., 2007), and thus may not be representative of actual aerosol acidity or pH (Guo et al., 2015; Hennigan et al., 2015; Murphy et al., 2017). In addition, ammonia in gas phase is also efficient at producing NOCs (Nguyen et al., 2012), which may play a complex role in the distribution of ammonium and NOCs in particulate phase. The formation of ammonium and NOCs would compete for ammonia, which may also potentially result in the negative correlation between the RPAs of NOCs and ammonium. Unfortunately, such a role remains unclear since the variations of ammonia were not available in the present study.This finding was consistent with the results discussed in section 3.1, indicating that particles containing a higher abundance of ammonium may not facilitate the formation of

NOCs.

Similarly, ambient observations reported from a forest site in Japan indicate that aerosol acidity likely plays an important role in the formation of WSON via acid-catalyzed reactions in summer (Miyazaki et al., 2014). Enhanced organic aerosol yields from gas-phase carbonyls in the acidic seed aerosol have been attributed to the occurrence of acid-catalyzed reactions (Jang et al., 2002). Furthermore, acidity could also play a significant role in the gas to particle partitioning of aldehydes (Herrmann et al., 2015; Liggio et al., 2005; Gen et al., 2018; De Haan et al., 2018; Kroll et al., 2005), although some studies have indicated that browning of some SOA occurs independently within a pH range of 4−10 (Nguyen et al.,

2012). Consistently higher relative acidity was observed for the internally mixed ammonium and NOCs particles, as compared to ammonium containing particles without NOCs (Fig.

S7).This may be due to the fact that the ammonium available to react with secondary oxidized organics was from the uptake of ammonia, regarding that NOCs were mainly supplied by heterogeneous reactions of oxidized organics, as discussed above. In this case, the formation of ammonium and NOCs would compete for ammonia, potentially resulting in a negative correlation between the RPAs of NOCs and ammonium as observed (Fig. 3).

A study shows that ammonia is more efficient at producing NOC than ammonium (Nguyen et al., 2012). The negative correlation between concentrations of WSON and $NH_4^-$ in filter samples (Fig. S7), may serve as quantitative support for the close association between

WSON formation and $NH_4^+$. Furthermore, the negative correlation between the RPA of

NOCs and ammonium, may indicate that the formation of NOCs is influenced by particle acidity, Consistently, the Nfs of ammonium that internally mixed with NOCs were inversely correlated with the RPAs of ammonium (Fig. S8). {Guo, 2015 #22779;Hennigan, 2015

**22780;Murphy, 2017 #22781}which is directly affected by the abundance of ammonium**

(as discussed in section 3.3). Consistently, the Nfs of ammonium that internally mixed with

NOCs were inversely correlated with the RPAs of ammonium (Fig. S8).

One may expect that NOCs were formed through the interactions between NOx and oxidized organics in gas phase followed by condensation (Fry et al., 2014; Stefenelli et al.,

2019; Lehtipalo et al., 2018). However, low correlation coefficients ($R^2$ = 0.02-0.13)

between NOCs and NOx indicates limited contribution of this pathways to the observed

NOCs. Also, NOCs formed through NOx and oxidized organics followed by partitioning would not be dependent on the amount of ammonium, which is incompatible with our results.

Multiple linear regression analysis was performed to predict the RPAs of NOCs generated from oxidized organics and ammonium, showing a close association ($R^2$ = 0.71,

$p < 0.01$) between the predicted RPAs and the observed values of NOCs (Fig. 4). Therefore, the interactions involving oxidized organics and ammonium may explain over half of the observed variations in NOCs in the atmosphere of Guangzhou. A fraction of the unaccounted

NOCs could be due to primary emissions and other formation pathways.

**3.2 Factors contributing to the NOCs resolved by positive matrix factorization (PMF)**

**analysis**

Figure 5 presents the PMF factor profiles obtained from the PMF model analysis (detailed information is provided in the SI) (Norris et al., 2009) and their diurnal variations. Around 75% of NOCs could be well explained by two factors, with 33% of the PMF resolved NOCs mainly associated with ammonium and carbonaceous ion peaks (ammonium factor), while 59% were mainly associated with oxidized organics (oxidized organics factor). The explained fraction of

NOCs by the ammonium and oxidized organic factors is consistent with the linear regression analysis. In addition, PMF analysis provided information on the factor contribution and diurnal variations, which may help explain the seasonal variations and processes of NOCs.

The ammonium factor showed a diurnal variation pattern peaking during early morning, which is consistent with the diurnal variation in RH (Zhang et al., 2019). This factor contributed to ~80% (Fig. S8) of the PMF resolved  NOCs during spring with  the highest RH  (Table S1), whereas  the oxidized organics factor dominated (> 80%) in  summer and fall. In winter, these two factors similarly contributed (~40%). This may indicate a potential role of aqueous pathways in the formation of NOCs, particularly during spring. Differently, the oxidized organics factor showed a pattern of diurnal variation, increasing from morning hours and peaking overnight, which may correspond to the photochemical production of oxidized organics and follow ed interactions with condensed ammonium. This pathway may explain the slightly late peaking of NOCs compared to oxidized organics, as  ammonium condensation is favorable overnight (Hu et al., 2008). While there were similarities in the fractions of oxidized organics in the oxalate factor and the oxidized organics factor, they only contributed to 8% of the PMF resolved modelled NOCs in the oxalate factor, which contained ~80% of the PMF resolvedmodelled oxalate. As previously discussed, these oxidized organics are also precursors for the formation of oxalate (Zhang et al., 2019).

Therefore, the PMF results  and therefore, these results suggest that there were are two competitive pathways for the evolution of these oxidized organics. Some oxidized organics formed from photochemical activities were further oxidized to oxalate, resulteding in a diurnal pattern of variation and with concentration peaks during the afternoon (Fig. 5), while others interact with $NH_3/NH_4^+$ ammonium to form NOCs, peaking during the nighttime.

However, the controlling factors for these pathways could not be determined in the present study. The unexplained NOCs (~25%) might be linked to the primary emissions, such as biomass burning (Desyaterik et al., 2013). It could be partly supported by the presence of potassium and various carbon ion clusters ($C_n^{+/-}$, n = 1, 2, 3, …) in the mass spectrum of

NOCs-containing particles (Fig. 1).

**3.2 3 Seasonal variations in the observed NOCs**

There is aA clear seasonal variation in of NOCs were also observed, with higher relative contributions during summer and autumn (Figs. 3 and 4), mainly due to the variations in oxidized organics and $NH_3/NH_4^+$. As discussed in section 3.3, particle acidity was lower

In this region, a larger contribution from secondary oxidized organics is typically observed during summer and autumn (Zhou et al., 2014; Yuan et al.,

2018). The seasonal maximum $NH_3$ concentrations have also been reported during the warmer seasons, corresponding to the peak emissions from agricultural activities and high temperatures, while the low $NH_3$ concentrations observed in colder seasons may be attributed to gas-to-particle conversion (Pan et al., 2018; Zheng et al., 2012). Such seasonal variation in NOCs  is also obtained in a model simulation, showing that the conversion of $NH_3$ into NOCs would result in a significantly higher reduction of gas-phase $NH_3$ during summer (67%) than winter (31%), due to the higher $NH_3$ and SOA concentrations present in the summer (Zhu et al., 2018). More primary NOCs may also be present during summer and autumn in the present study, due to the additional biomass burning activities in these seasons (Chen et al., 2018; Zhang et al., 2013).

The seasonal variations  of NOCs can be adequately explained by the variations in concentrations of oxidized organics and ammonium (Fig. 4), although the hourly variations during each season  are not well explained, as indicated by the lower

$R^2$ values (Table S2). The correlation coefficients ($R^2$) range from 0.24 to 0.57 for inter- seasonal variations the regressions were found to be significant.

variation of NOCs was dependent on seasons, despite the correlations between NOCs and oxidized

During spring,

[revised manuscript text omitted]

---

## Author Comment (AC2) · 24 Oct 2019

**Response to comments**

This paper analyzed single particle aerosol mass spectrometer (SPAMS) data for ambient aerosols and found there are relations between CN-/CNO- ion intensities and some other species, such as oxidized organic ions and ammonium. It is an interesting report. But there are some concerns which need to be addressed before publication.

We would like to thank the reviewer for his/her useful comments and recommendations to improve the manuscript. We have addressed the specific comments in the sections below and made the appropriate revisions to the manuscript. Reviewer comments are in black text followed by our response in blue text.

One of the major problems is that this paper attribute oxidized organics to secondary formation. However, it may not be the case. Biomass burning or coal combustion can also produce oxidized organics including large amounts of NOCs. Actually, in many previous single particle mass spectrometry studies, CN- and CNO- were taken as ion markers for combustion sources. The authors need to provide more evidences either to rule out the possibility of primary oxygenated organics and primary NOCs or to distinguish the secondary organics from the primary ones.

Thanks for the suggestion. In our manuscript, oxidized organics, represented as formate at m/z -45 $[HCO_2]^-$, acetate at m/z -59 $[CH_3CO_2]^-$, methylglyoxal at m/z -71 $[C_3H_3O_2]^-$, glyoxylate at m/z -73 $[C_2HO_3]^-$, pyruvate at m/z -87 $[C_3H_3O_3]^-$, malonate at m/z -103 $[C_3H_3O_4]^-$ and succinate at m/z -117 $[C_4H_5O_4]^-$, which are generally regarded as secondary compositions (Zhang et al., 2017; Zauscher et al., 2013; Lee et al., 2003). To make it clear, we revised the original description to "These oxidized organics showed their pronounced diurnal trends with afternoon maximum, and were highly correlated (r = 0.72 - 0.94, p < 0.01) with each other. Therefore, they were primarily attributed to the secondary oxidized organics from photochemical oxidation products of various volatile organic compounds (VOCs) (Paulot et al., 2011; Zhao et al., 2012; Ho et al., 2011), and the details can be found in our previous publication (Zhang et al., 2019).". Please refer to Lines 136-141 of the revised manuscript.

We strongly agree with the reviewer that biomass burning or coal combustion can also produce oxidized organics and NOCs. As discussed above, these oxidized organics most probably formed from secondary process. In the original manuscript, we provided evidence for the secondary formation of NOCs. However, the primary NOCs cannot be ruled out. In the revised manuscript (line 271-274), we have included the following sentence to mention this: "The unexplained NOCs (~25%) might be linked to the primary emissions, such as biomass burning (Desyaterik et al., 2013). It could be partly supported by the presence of potassium and various carbon ion clusters ($C_n^{+/-}$, n = 1, 2, 3, …) in the mass spectrum of NOCs-containing particles (Fig. 1).".

Another major concern is that how well CN-/CNO- ions can represent total NOCs. Can they represent 25%, 50% or 75% of total NOCs? The paper needs to provide more discussion on this issue.

Thanks for the comment. We understand that it would be better if the exact fraction of NOCs represented by $CN^-$/$CNO^-$ can be obtained. Unfortunately, how well $[CN]^-$ / $[CNO]^-$ ions could represent NOCs cannot be quantified, although they were the most commonly reported NOCs peaks by single particle mass spectrometry (Silva and Prather, 2000; Zawadowicz et al., 2017; Pagels et al., 2013). In the present study, $[CN]^-$ / $[CNO]^-$ ions are among the major peaks detected by the SPAMS (Fig. 1). A rough estimate from the peak area ratio of $[CN]^-$ / $[CNO]^-$ ions and the most likely NOCs fragments (i.e., various amines, and an entire series of nitrogen-containing cluster ions $C_nN^-$, n = 1, 2, 3, …) (Silva and Prather, 2000) shows that $[CN]^-$ / $[CNO]^-$ ions may represent more than 90% of these NOCs peaks. It has been added in section 2.2.

Pagels, J., Dutcher, D. D., Stolzenburg, M. R., McMurry, P. H., Galli, M. E., and Gross, D. S.: Fine-particle emissions from solid biofuel combustion studied with single-particle mass spectrometry: Identification of markers for organics, soot, and ash components, J. Geophys. Res.-Atmos., 118, 859-870, doi:10.1029/2012jd018389, 2013.

Silva, P. J., and Prather, K. A.: Interpretation of mass spectra from organic compounds in aerosol time-of-flight mass spectrometry, Anal. Chem., 72, 3553-3562, 2000.

Zawadowicz, M. A., Froyd, K. D., Murphy, D. M., and Cziczo, D. J.: Improved identification of primary biological aerosol particles using single-particle mass spectrometry, Atmos. Chem. Phys., 17, 7193-7212, doi:10.5194/acp-17-7193-2017, 2017.

The third concern is that ammonium sulfate is very difficult to be ionized under 266 nm UV laser. Thus, it is likely that some mass spectra of particles do not contain NH4+ peak but these particles may still contain ammonium sulfate. The authors also need to provide some discussions on this possibility.

Thanks for the comment. It is true that pure ammonium sulfate is very difficult to be ionized under 266 nm UV laser used in our study. In the present study, this may not be the case since we focused on the NOCs-containing particles, in which the Nfs of ammonium varied in a wide range (~40-90%) (Fig. 2). Such possibility has been added in Lines 27-28 of the revised *Supplements*.

Specific comments:

Line 54: how much is "large"? It would be always better to provide a number or range.

Thanks for the comment. We have revised the sentence to "Nitrogen-containing organic compounds (NOCs) substantially contribute to the pool of BrC". And we have also stated that "The particulate organic nitrogen accounts for a large fraction of total airborne nitrogen (~30%)". Please refer to Lines 57-61 of the revised manuscript.

Line 149: "so on" is a bit informal. I would change "so on" to "so forth"

It has been revised as suggested.

Line 220: How do you come up with this statement: ": : :explain over half of the observed variations in NOCs in the atmosphere of Guangzhou."? Please elaborate and provide more details.

Thanks for the comment. Multiple linear regression analysis was performed to predict the RPAs of NOCs generated from oxidized organics and ammonium, showing a close association ($R^2 = 0.71$, $p < 0.01$) between the predicted RPAs and the observed values of NOCs (Fig. 4). Based on this result, we infer that over half of the observed variations of NOCs can be explained by the interactions involving oxidized organics and ammonium. This is also supported by the PMF analysis provided in Fig. 5. The sentence has been revised to "The result indicates that interactions involving oxidized organics and ammonium could explain over half of the observed variations in NOCs in the atmosphere of Guangzhou.", and the discussion can be found in Lines 192-200 of the revised manuscript.

Line 224: Please report if the PMF analysis reaches convergence or not. How much is the error of the PMF modelling in the paper?

Thanks for the comment. Such information has been added in the *Supplements*. It can be found in section "Positive matrix factorization analysis", as "PMF solutions with 2−5 factors were tested and showed convergence results. The relevant Q values and $Q_{robust}$ / $Q_{theory}$ for these solutions are shown in Table S3.", and "An uncertainty of 50% in RPA was used due to the shot-to-shot fluctuations of desorption laser and complex particle matrix (Zauscher et al., 2013)."

Line 387: Check English

Thanks for the comment. We have carefully checked and corrected the syntax errors.

**High secondary formation of nitrogen-containing organics (NOCs) and its**

**possible link to oxidized organics and ammonium**

[revised manuscript text omitted]

Similarly, ambient observations reported from a forest site in Japan indicate that aerosol
acidity likely plays an important role in the formation of WSON via acid-catalyzed reactions
in summer (Miyazaki et al., 2014). Enhanced organic aerosol yields from gas-phase
carbonyls in the acidic seed aerosol have been attributed to the occurrence of acid-catalyzed
reactions (Jang et al., 2002). Furthermore, acidity could also play a significant role in the
gas to particle partitioning of aldehydes (Herrmann et al., 2015; Liggio et al., 2005; Gen et
al., 2018; De Haan et al., 2018; Kroll et al., 2005), although some studies have indicated that
browning of some SOA occurs independently within a pH range of 4–10 (Nguyen et al.,
2012). Consistently higher relative acidity was observed for the internally mixed ammonium
and NOCs particles, as compared to ammonium containing particles without NOCs (Fig.
S7). This may be due to the fact that the ammonium available to react with secondary
oxidized organics was from the uptake of ammonia, regarding that NOCs were mainly
supplied by heterogeneous reactions of oxidized organics, as discussed above. In this case,
the formation of ammonium and NOCs would compete for ammonia, potentially resulting
in a negative correlation between the RPAs of NOCs and ammonium as observed (Fig. 3).
A study shows that ammonia is more efficient at producing NOC than ammonium (Nguyen
et al., 2012). The negative correlation between concentrations of WSON and $NH_4^-$ in filter
samples (Fig. S7), may serve as quantitative support for the close association between

WSON formation and $NH_4^+$. Furthermore, the negative correlation between the RPA of

NOCs and ammonium, may indicate that the formation of NOCs is influenced by particle acidity, Consistently, the Nfs of ammonium that internally mixed with NOCs were inversely correlated with the RPAs of ammonium (Fig. S8). {Guo, 2015 #22779;Hennigan, 2015

#22780;Murphy, 2017 #22781}which is directly affected by the abundance of ammonium (as discussed in section 3.3). Consistently, the Nfs of ammonium that internally mixed with

NOCs were inversely correlated with the RPAs of ammonium (Fig. S8).

One may expect that NOCs were formed through the interactions between NOx and oxidized organics in gas phase followed by condensation (Fry et al., 2014; Stefenelli et al.,

2019; Lehtipalo et al., 2018). However, low correlation coefficients ($R^2$ = 0.02-0.13)

between NOCs and NOx indicates limited contribution of this pathways to the observed

NOCs. Also, NOCs formed through NOx and oxidized organics followed by partitioning would not be dependent on the amount of ammonium, which is incompatible with our results.

Multiple linear regression analysis was performed to predict the RPAs of NOCs generated from oxidized organics and ammonium, showing a close association ($R^2$ = 0.71,

$p < 0.01$) between the predicted RPAs and the observed values of NOCs (Fig. 4). Therefore, the interactions involving oxidized organics and ammonium may explain over half of the observed variations in NOCs in the atmosphere of Guangzhou. A fraction of the unaccounted

NOCs could be due to primary emissions and other formation pathways.

[revised manuscript text omitted]

---

## Referee Report (RR1)

Overall, the authors have done well to address some of the referee comments, but not all. There are several major issues that remain, which prevent me from endorsing the manuscript for publication. For the comments that have not been addressed sufficiently, I have the original referee comment below in quotes, followed by my assessment of the changes and additional comments.

Original comment: "For $NH_3$, a positive correlation is observed between the number fraction of particles with $NH_4^+$ and NOC, while a negative correlation is observed between the relative peak areas of these compound classes. There is not a reasonable explanation given for this surprising and apparently contradictory behavior."

New comment: The authors' explanation for this observation still does not make sense. The authors state that ammonium is necessary for NOC formation, but that ammonium also inhibits NOC formation. Various factors contributing to this counterintuitive result are hypothesized (e.g., acidity, competition for gas-phase ammonia), but none are consistent with the results in Figure 3c and 3d. The discussion in the revised manuscript (revised manuscript with track changes lines 205 - 214) actually contradicts their finding, since ammonium does not inhibit NOC formation in any of the references cited in this added paragraph. Based on the other referee's comments, it seems there is a possibility that the results in Figures 3c and 3d stem from a measurement artifact (ionization efficiency changing with composition), rather than an actual physical/chemical process occurring in the atmosphere. The authors need to present a detailed, logical argument for this observation.

Original comment: "NOx is completely ruled out as a contributor to NOC formation on the basis of poor (or no) correlations between NOC and NOx. However, this is a misinterpretation of the data. Many factors (different removal processes and lifetimes of particles vs. gasses, primary vs. secondary species, etc.) could contribute to a lack of correlation even if NOx did contribute to NOC formation."

New comment: First of all, the authors used the referee's comment word-for-word in their revised manuscript (lines 460-462 in the track changes version). This is inappropriate, and constitutes plagiarism of this sentence. Clearly, this should be changed.

The above issue aside, I still do not believe this comment was adequately addressed in the revision. There is too much emphasis on simple linear correlations, when that is not expected for the chemistry in this system. For example, NOx controls the branching of VOC reactions, which will in turn affect product distributions, including NOCs; however, this will not (in most cases) result in a linear correlation between NOx and NOCs. Likewise, NOx affects nitrate radical formation, which can form NOCs, but a linear relationship between NOx and NOCs will not necessarily occur even if this is the predominant pathway for NOC production. Further, the references cited in lines 451-453 (track changes version) to support their position do not show linear correlations between NOx and NOC formation in systems representative of a polluted

urban atmosphere.  Therefore, the possible role of NOx in NOC formation is not accurately described in the manuscript revision.

Finally, the revised manuscript needs editing for grammar and language.

---

## Author Response (AR2)

**Response to comments 1**

Review of the manuscript revision by G. Zhang et al.

Overall, the authors have done well to address some of the referee comments, but not all. There are several major issues that remain, which prevent me from endorsing the manuscript for publication. For the comments that have not been addressed sufficiently, I have the original referee comment below in quotes, followed by my assessment of the changes and additional comments.

We would like to thank the referee again for his/her valuable comments to further improve the manuscript. We have addressed the specific comments in the sections below and made the appropriate revisions to the manuscript. The referee's comments are in the black text followed by our response in blue text.

Original comment: "For $NH_3$, a positive correlation is observed between the number fraction of particles with NH4+ and NOC, while a negative correlation is observed between the relative peak areas of these compound classes. There is not a reasonable explanation given for this surprising and apparently contradictory behavior."

New comment: The authors' explanation for this observation still does not make sense. The authors state that ammonium is necessary for NOC formation, but that ammonium also inhibits NOC formation. Various factors contributing to this counterintuitive result are hypothesized (e.g., acidity, competition for gas-phase ammonia), but none are consistent with the results in Figure 3c and 3d. The discussion in the revised manuscript (revised manuscript with track changes lines 205 - 214) actually contradicts their finding, since ammonium does not inhibit NOC formation in any of the references cited in this added paragraph. Based on the other referee's comments, it seems there is a possibility that the results in Figures 3c and 3d stem from a measurement artifact (ionization efficiency changing with composition), rather than an actual physical/chemical process occurring in the atmosphere. The authors need to present a detailed, logical argument for this observation.

Thanks for pointing this out. Regarding the referee's concern, we also agree with the comment that "ammonium also inhibits NOC formation" is not precise enough to pinpoint the observation results in Figure 3. It has been revised to "The formation of NOCs requires the presence of ammonium, but is most probably influenced by the relative amount of ammonium in individual particles" in the revised manuscript, which is believed to be more appropriate.

We understand that the reviewer still considered this observation as a counterintuitive result. As previously replied, we highlighted the contradictory correlation between the hourly detected number and RPAs of NOCs and ammonium as a surprising and interesting result. We think that the referee's confusion maybe mainly due to our unclear expressions on this issue. As replied to the referee 2#, this is unlikely due to measurement artifact. To make it clearer, we attempted to provide more discussion in the revised manuscript.

First, from the observational results in Figure 3d, we stated that the formation of NOCs requires the involvement of ammonium. It is easy to understand that ammonium is important in the formation of NOCs, from the high correlations between their hourly detected number. Consistently, NOCs and ammonium showed high internal mixing state in Figure 2.

Second, we provided additional explanations on the inverse correlation between RPAs of NOCs and ammonium as shown in Figure 3c. The negative correlation between their RPAs signifies that the formation of NOCs is most probably influenced by the relative amount of ammonium in individual particles. This is supported by our data, both from filter samples and individual particle analysis. There is a negative correlation between concentrations of WSON and $NH_4^+$ for the filter samples (Fig. S6). It can be seen from Fig. S7 that lower RPAs of ammonium correspond to higher Nfs of ammonium that internally mixed with NOCs. Such an inverse correlation could also serve as evidence to explain the influence of the relative amount of ammonium on the formation of NOCs.

Furthermore, we discussed the theoretical possibility that the formation of NOCs is influenced by the relative amount of ammonium. We showed that this is theoretically possible since the formation of NOCs may be enhanced by particle acidity (Miyazaki et al., 2014; Aiona et al., 2017; Nguyen et al., 2012), which is substantially affected by the amount of ammonium. This could also be supported by our data, as shown in Figure S6, that the internally mixed ammonium and NOCs particles had higher relative acidity than the ammonium-containing particles without NOCs. It is also noted that particle acidity could play a significant role in the gas-to-particle partitioning of aldehydes (Herrmann et al., 2015; Liggio et al., 2005; Gen et al., 2018; De Haan et al., 2018; Kroll et al., 2005), which are precursors for the formation of the oxidized organics.

Original comment: "NOx is completely ruled out as a contributor to NOC formation on the basis of poor (or no) correlations between NOC and NOx. However, this is a misinterpretation of the data. Many factors (different removal processes and lifetimes of particles vs. gasses, primary vs. secondary species, etc.) could contribute to a lack of correlation even if NOx did contribute to NOC formation."

New comment: First of all, the authors used the referee's comment word-for-word in their revised manuscript (lines 460-462 in the track changes version). This is inappropriate, and constitutes plagiarism of this sentence. Clearly, this should be changed.

Thanks for pointing this out. We have changed the sentence to "A lack of correlation could probably be attributed to that NOx affects the formation of NOCs in various ways (e.g., peroxy radical chemistry in VOCs oxidation mechanisms and formation of nitrate radicals) (Xu et al., 2015; Zhang et al., 2018), and thus may not linearly contribute to the formation of NOCs."

The above issue aside, I still do not believe this comment was adequately addressed in the revision. There is too much emphasis on simple linear correlations, when that is not expected for the chemistry in this system. For example, NOx controls the branching of VOC reactions, which will in turn affect product distributions, including NOCs; however, this will not (in most cases) result in a linear correlation between NOx and NOCs. Likewise, NOx affects nitrate radical formation, which can form NOCs, but a linear relationship between NOx and NOCs will not necessarily occur even if this is the predominant pathway for NOC production. Further, the references cited in lines 451-453 (track changes version) to support their position do not show linear correlations between NOx and NOC formation in systems representative of a polluted urban atmosphere. Therefore, the possible role of NOx in NOC formation is not accurately described in the manuscript revision.

Thanks for the referee's constructive comment. It greatly helps us to refine our discussion on the possible contribution of NOx pathways to the formation of NOCs. We agree with the comment that simple linear correlations may not be expected for the chemistry involving NOx and NOCs. This is because that NOx affects the formation of NOCs in various ways (e.g., peroxy radical chemistry in VOCs oxidation mechanisms and formation of nitrate radicals) (Xu et al., 2015; Zhang et al., 2018). Besides, the related products or intermediates were not available to elucidate such complex chemical processes in the present study. Therefore in the revised manuscript, we mentioned that NOx may play a certain role in the conversion of oxidized organics to NOCs, and yet it cannot be quantified in the present study. This is supported by the relationship between peak ratios of NOCs/oxidized organics and NOx (Fig. 6), showing that the ratios generally increase with an increasing level of NOx ($R^2 = 0.18$, $p < 0.01$). It is noted that low correlation coefficients ($R^2 = 0.02 – 0.13$) between NOx and NOCs might not indicate a limited contribution of NOx to the formation of NOCs. A lack of correlation could probably be interpreted by the fact that NOx affects the formation of NOCs in various ways (e.g., peroxy radical chemistry in VOCs oxidation mechanisms and formation of nitrate radicals) (Xu et al., 2015; Zhang et al., 2018), and thus may not linearly contribute to the formation of NOCs.

Finally, the revised manuscript needs editing for grammar and language.

Thanks for the comment. We have carefully examined and corrected possible technical errors.

[revised manuscript text omitted]

---

## Author Response (AR3)

EDITORIAL SUGGESTIONS:

*1. I would suggest mentioning in the introduction that there are several types of NOC in particulate matter: primary NOC formed by direct emissions from various sources such as biomass burning; organic nitrates and nitroaromatic compounds formed by the traditional gas-phase oxidation of VOCs, and believed to represent the major part of NOC; heterocyclic NOC compounds formed by reactions of carbonyls with ammonia and amines. The contribution of the last group to NOC is still uncertain but as described in some of the reviews cited in this paper they are likely to be minor in abundance. Right now, the most important type of NOC is mentioned in passing on lines 80-83, which is not ideal. I think distinguishing various types of NOC more clearly in the introduction will make it easier to discuss the results.*

We would like to thank the editor for the valuable comments to further improve our manuscript. We have revised the *Introduction* according to the suggestion. To distinguish various types of NOCs makes the discussion on the origin and formation pathways clearer. We also note that secondary NOCs, such as organic nitrates and nitroaromatic compounds, are believed to be mainly formed in the gas-phase by interaction between volatile organic compounds and oxidations. However, reactions involving mixtures of atmospheric aldehydes (e.g., methylglyoxal/glyoxal) and ammonium/amines have not been confirmed with ambient data and the relative contribution of heterocyclic NOCs is still uncertain, although they are likely to be minor (at a level of several ng m$^{-3}$) in abundance.

*2. I find the references do not always directly support the statements. I encourage the authors to go through the reference list one more time carefully and update them as needed.*

Thanks for the comment. We have carefully examined and updated the references.

*Examples of references that do not correlate well with the statements:*

*Line 59 and line 67: Feng et al. (2013) does not mention any nitrogen containing organic compounds in their paper, so this reference does not support the statements on these lines. Feng paper is about the importance of BrC in general, not about NOC in BrC.*

Thanks for the comment. The reference has been removed.

*Line 60: Reference to Noziere et al. (2015) is the only one in this list that is not a review. It could be a better reference for supporting the previous statement about NOC in BrC.*

Thanks for the comment. The reference has been removed.

*Line 79: Mang et al. (2008) paper did not conclude that $NH_3$ reactions lead to the formation of NOC. The authors may instead refer to Bones et al. (2010) paper, which they already cite.*

Thanks for the comment. The reference has been revised to (Huang et al., 2017) :

*Huang, M., Xu, J., Cai, S., Liu, X., Zhao, W., Hu, C., Gu, X., Fang, L., and Zhang, W.: Characterization of brown carbon constituents of benzene secondary organic aerosol aged with ammonia, J. Atmos. Chem., 75, 205-218, doi:10.1007/s10874-017-9372-x, 2017.*

*Line 83 and 324: Paper by Lehtipalo et al. (2018) is not the best choice to support the importance of NOC formation through gas-phase mechanisms. The classic organonitrate formation in photooxidation of hydrocarbons is described in many old papers and books*

Thanks for the comment. This reference has been removed accordingly. We think the following two references may be more appropriate.

*Seinfeld, J. H., and Pandis, S. N.: Atmospheric Chemistry and Physics: From Air Pollution to Climate Change, edited by: John Wiley&Sons, I., John Wiley&Sons, Inc., New Jersey, 2006.*

*Ziemann, P. J., and Atkinson, R.: Kinetics, products, and mechanisms of secondary organic aerosol formation, Chem. Soc. Rev., 41, 6582-6605, doi:10.1039/c2cs35122f, 2012.*

*Lin 98 and 226: Aiona et al. (2017) paper did not study effects of pH or RH on NOC formation. It looked the photodegradation of NOC once they formed. Perhaps a different paper should be cited here.*

Thanks for the comment. This reference has been removed accordingly.

*Line 239: Pure gas-phase formation of NOC was also demonstrated, for example, in [Duporte et al. 2017, Chemical Characterization of Gas- and Particle-Phase Products from the Ozonolysis of alpha-Pinene in the Presence of Dimethylamine. Environ. Sci. Technol. 2017, 51, (10), 5602-5610.]*

Thanks for the comment. We think that this reference is not related to our discussion on the distribution of ammonium and NOCs in the particulate phase, and thus it is not included.

*TECHNICAL CORRECTIONS:*

*Line 56: (BrC), capable of a comparable level of light absorption-> (BrC). BrC has a comparable level of light absorption*

*Line 107: Sampling was constructed -> Sampling was done*

*Line 120: here and below "m/z" should be italicized*

*Line 136: distributing along their vacuum -> as a function of their vacuum*

*Line 140: 39 -> m/z 39*

*Line 136, 139, 142, 279, 760: NOCs-containing -> NOC-containing*

*Line 149: photochemical oxidation products of -> photochemical oxidation of*

*Line 225: enhanced -> affected*

*Line 332: unformatted citations*

*Line 409, 451, 607, 742: sub and superscripts in chemical formulas*

*Line 417: missing volume and page*

*Line 551: verify page number*

*Lin e576: missing page number*

*Figure 3 may benefit from a higher resolution*

*Figure 6 contains faint gridlines in the PDF version that should be removed. Figure 6 caption only mentions dependence on RH but dependence of NOx is shown also. The caption needs to be amended.*

Thanks for pointing these out. We have corrected them accordingly.

*SUPPORTING INFORMATION:*

*I would recommend placing figures and table immediately next to whether they are mentioned and not at the end of the document.*

*Line 44: 14 -> Fourteen*

*Line 44: -97[HSO4]- -> m/z -97 [HSO4]- and similar edits in this paragraph*

*Line 64, 66: dependent -> dependence*

*Line 89: missing space before the Pratt reference*

*Line 113: there are earlier reports of making BrC by methyl glyoxal + AS reaction, I would use the original report.*

*Lin 159: Acs -> ACS*

*Figure S1 has low image quality*

*Figure S2: In my opinion, this figure should be split into 4 panels, one for each season*

*Figure S5: the traces in the left panels are very hard to see*

Thanks for pointing these out. We have addressed these issues accordingly.

[revised manuscript text omitted]